# Surgical Treatment Following Failed Medical Treatment of an Interstitial Pregnancy

**DOI:** 10.3390/medicina58070937

**Published:** 2022-07-15

**Authors:** Stefano Restaino, Elena De Gennaro, Stefano Floris, Guglielmo Stabile, Giulia Zinicola, Felice Sorrentino, Giuseppe Vizzielli, Lorenza Driul

**Affiliations:** 1Obstetrics and Gynecology Unit, Department of Obstetrics, Gynecology and Pediatrics, Department of Medical Area DAME, Udine University Hospital, 33100 Udine, Italy; stefano.restaino@asufc.sanita.fvg.it (S.R.); stefano.floris@asufc.sanita.fvg.it (S.F.); 2Clinic of Obstetrics and Gynecology, Department of Medical Area (DAME), Hospital of Udine, University of Udine, 33100 Udine, Italy; degennaro.elena@spes.uniud.it (E.D.G.); giuseppe.vizzielli@uniud.it (G.V.); lorenza.driul@uniud.it (L.D.); 3Institute for Maternal and Child Health IRCCS “Burlo Garofolo”, 34100 Trieste, Italy; guglielmost@gmail.com; 4Department of Medicine, Surgery and Health Sciences, University of Trieste, 34100 Trieste, Italy; giulia.zinicola@burlo.trieste.it; 5Department of Medical and Surgical Sciences, University of Foggia, 71100 Foggia, Italy

**Keywords:** interstitial pregnancy, medical treatment, laparoscopy, salpingectomy, mifepristone, methotrexate

## Abstract

Interstitial pregnancy (IP) is a type of ectopic pregnancy in which the embryo implants in the interstitial part of the Fallopian tube. It accounts for 2% of all ectopic pregnancies. Signs and symptoms appear later than the other forms of ectopic pregnancies because of its peculiar location. The gold standard for its diagnosis is transvaginal ultrasound. The treatment can be medical or surgical. Medical treatment is based on the systemic or local injection of methotrexate (MTX); a dose of mifepristone can be added with a reported 85–90% success rate. The surgical option is laparoscopic unilateral cornuostomy or unilateral salpingectomy. The therapeutic choice is based on symptoms, serum β-human chorionic gonadotropin (β-hCG) values, and sonographic features. Furthermore, the patient’s fertility perspectives should be considered. We report a case of IP in a Caucasian woman of 29 years old, with a previous salpingectomy for ectopic pregnancy medically treated by a double dose of intramuscular MTX 50 mg/m^2^ combined with a single dose of leucovorin 15 mg and a single dose of mifepristone 600 mg orally. Medical therapy failed as suggested by the sudden onset of intense pelvic pain after 10 days. Because of the clinical symptoms and the sonographic suspicious of pregnancy rupture due to the modest amount of fluid in the pouch of Douglas, clinicians decided on an urgent unilateral laparoscopic salpingectomy. The hemoperitoneum was drained. The patient was discharged two days later and β-hCG serum levels became negative after 45 days. The advantages of fertility sparing should be weighted according to the patient’s reproductive perspectives. Appropriate counseling is therefore key in managing the treatment of interstitial pregnancy.

## 1. Introduction

Interstitial pregnancy (IP) is a rare condition representing about 2–6% of all ectopic pregnancies [1]. It occurs when the gestational sac implants into the interstitial portion of the Fallopian tube, where it penetrates the myometrium [2,3]. Maternal mortality rate related to IP is around 2–2.5% [4]. Symptoms include vaginal bleeding and/or lower abdominal pain, often developing at later gestational ages than in other types of ectopic pregnancy. That is due to the increased resistance of the interstitium, which allows the gestational sac to develop for longer than in the tubes before rupture [1]. The gold standard for the diagnosis of interstitial pregnancy is transvaginal ultrasound. The imaging shows an empty uterus with a inhomogeneous, hypoechoic, and highly vascularized mass dislocated laterally into the interstitial portion of the Fallopian tube [3]. A gestational sac or an embryo with cardiac activity can be visualized. Peculiar features are the interstitial rail sign, an echoic line connecting the endometrium to the pregnancy, and myometrial thickness less than 5 mm [3]. Serum β-human chorionic gonadotropin (β-hCG) is lower than in a uterine pregnancy and it can reach a plateau. Treatment is based on the value of β-hCG and sonographic features. Medical or surgical strategies are possible. Medical therapy is based on methotrexate (MTX), injected locally or systemically, in single or multiple dose regimen [5]. An oral dose of mifepristone can be added [6]. Medical therapy must be administered only to hemodynamically stable patients and its success should be assessed by the decline of serum β-hCG, which occurs in about 85–90% of cases [7]. Surgical treatment is recommended for patients who do not respond to medical therapy, who are hemodynamically unstable, or who have clinical or sonographic features such as high levels of β-hCG or a large gestational sac. The most effective surgical treatments are laparoscopic unilateral cornuotomy and laparoscopic unilateral salpingectomy [1,8]. Each patient must be counselled on the best therapeutic options, depending on their past medical history and future reproductive perspectives. Because of the rarity of the disease there is limited evidence on which is the best option for a safer management, and the evidence is mostly based on small case series reported in the literature. We would like to contribute to the existing literature with our experience. More data are necessary to be able to outline the best therapeutic option for this rare and life-threating condition in the future.

## 2. Case Presentation

We report a case of IP in a 29-year-old Caucasian woman (gravidity 3, parity 1) with a history of unilateral salpingectomy for tubal ectopic pregnancy, after a failed medical treatment. In this instance, she was medically treated with a double dose of intramuscular MTX 50 mg/m^2^ combined with a single dose of leucovorin 15 mg and a single dose of mifepristone 600 mg orally. She presented to the emergency room with 6 weeks of amenorrhea, reporting lower abdominal pain and vaginal bleeding. Physical examination revealed a tenderness in the lower right quadrant; Blumberg sign was negative. Transvaginal ultrasound examination was performed by an expert operator, and it was showed a gestational sac of mm 1.3 × 0.9 in size containing an embryo of mm 0.57 with cardiac activity, localized outside the uterus, in the right interstitium (Figure 1 and Figure 2).

Interstitial rail sign and a reduced thickness of the myometrium around the gestational sac were visualized. No sign of hemoperitoneum was revealed. Serum level of β-human chorionic gonadotropin (β-hCG) was 5520 mUI/mL. No further investigations were performed. The patient was then admitted to the gynecology unit with the diagnosis of right interstitial pregnancy. Counselling about therapeutic strategies, risks and benefits of the medical and the surgical treatments, as well as patient’s future fertility perspectives was performed. Due to the past obstetrical history (a previous tubal ectopic pregnancy treated by salpingectomy), ultrasound features, hemodynamic stability, and the patient’s desire of future pregnancies, the medical treatment was chosen. A double dose of intramuscular MTX 50 mg/m^2^ was administrated, along with a single dose of leucovorin 15 mg and a single dose of mifepristone 600 mg orally. The patient remained asymptomatic for all 10 days of hospitalization. Over the days, ultrasound examinations showed the gestational sac progressively reducing in size, with embryo cardiac activity absence. The patient was therefore discharged with a prescription of serial β-hCG serum level and sonographic longitudinal follow-up, weekly. Two days after the discharge, the woman came back to the emergency room because of the onset of an intense, stabbing pelvic pain. Transvaginal ultrasound revealed free fluid in the pouch of Douglas. β-hCG serum level was of 617 mUI/mL. Because of the clinical hemodynamical instability, gynecologists decided for a surgical strategy and the patient underwent urgent right laparoscopic salpingectomy with removal of the interstitial pregnancy. A total of 1000 cc of hemoperitoneum was drained (Figure 3). The patient was discharged two days later. Follow-up was done 40 days after surgery: physical and transvaginal ultrasound examination were regular and β-hCG serum levels became negative after 45 days.

## 3. Discussion

Interstitial pregnancy (IP) is a rare condition. Diagnosis of IP is based on clinical features, such as lower abdominal pain and vaginal bleeding, serum β-hCG level, and sonographic criteria. Serum β-hCG level trends can be helpful to distinguish an intrauterine pregnancy from an ectopic pregnancy [1]. Transvaginal ultrasound examination allows the diagnosis of this rare condition when the following criteria are present: an empty uterine cavity, a gestational sac located into the interstitial portion of the tube and surrounded by a thin myometrium, the interstitial rail sign (an echoic line connecting the pregnancy to the uterine cavity), and increased vascularization at Color–Doppler examination [9]. Diagnosis of IP is becoming increasingly timely due to the better accuracy of the latest ultrasound systems [10]. Magnetic resonance imaging may be used if ultrasound is inconclusive, especially to differentiate it from an angular or intrauterine pregnancy [11]. In our case, ultrasound was performed by an expert operator and was conclusive for interstitial ectopic pregnancy. No further investigations were necessary. Thanks to early diagnosis, instances of life-threatening emergencies due to pregnancy ruptures are decreasing [10]. It has been reported that non-surgical treatment for interstitial pregnancy has a success rate of about 85 to 90% of cases [6,12]. The choice of medical treatment can be considered in case of hemodynamic stability of the patient and for fertility preservation purposes; other factors that should be taken into consideration are the size of the gestational sac and thickness of the surrounding myometrium [6]. Despite the rarity of the disease, there are various treatment options, conservative and radical. The established medical therapy is based on the intramuscular injection of methotrexate, a chemotherapy drug used in ectopic pregnancies in single or multiple dose protocols [3,13]. According to the Stovall protocol, an intramuscular single-dose of MTX (50 mg/m^2^) is administered, followed by serum β-hCG check. A progressive decrease of serum β-hCG levels of at least 15% is expected between day 4 and day 7 after the injection [14]. If the decrease is less than 15%, a second dose of MTX is required [10]. The multiple dose protocol consists of alternating intramuscular injections of 1 mg/kg of MTX on days 1, 3, 5, 7 and 0.1 mg/kg of leucovorin on days 2, 4, 6, 8. The protocol should be continued until the β-hCG level drops 15% from its previous measurement [4,13]. An alternative protocol involves the local injection of MTX into the gestational sac. Halperin et al. demonstrated that the combined local sonographically guided and systemic injection of methotrexate is associated with a successful outcome in asymptomatic patients presenting with ectopic pregnancy and fetal cardiac activity [15]. However, the systemic route appears to be safer because of the high vascularization of the gestational sac [6]. The addition of a single dose of mifepristone was shown to improve success rates of the medical treatment [11,16]. Mifepristone is a competitive progesterone receptor antagonist, exerting its action through the detachment of the trophoblast from the uterine decidua, the degeneration of the corpus luteum, and the induction of uterine contractions. MTX and mifepristone act in synergy by promoting the lysis of trophoblast cells [17]. Considering clinical and sonographic findings, previous contralateral salpingectomy for tubal ectopic pregnancy and the strong desire to preserve her fertility, we choose a conservative medical approach for our patient. An accurate counselling was made with regards to the treatment-related risks. Despite a double dose protocol of MTX combined with mifepristone, a β-hCG value of 5520 mU/mL at the diagnosis, and the absence of cardiac activity of the embryo at ultrasound examination during the follow-up, the pregnancy continued to evolve. According to the study of Barnhart et al., the multiple dose protocol seems to be more effective than the single dose protocol [18]. However, according to Conti et al., the multiple dose protocol leads to a higher number of side effects such as bone marrow suppression and granulocytopenia [19]. We retained the safer systemic route than intralesional injection of MTX. We chose a double dose of MTX combined with mifepristone to increase the effectiveness of medical treatment and to prevent possible adverse events related to multiple doses of MTX. Signs and symptoms of rupture with evidence of hemoperitoneum appeared just 10 days after the treatment, even though the size of the gestational sac at ultrasound and the levels of β-hCG were decreasing (617 mUI/mL). To our knowledge, this is the first reported case of failure of combined medical therapy with MTX and mifepristone for the treatment of an interstitial pregnancy and subsequent urgent surgical treatment due to the appearance of signs and symptoms of pregnancy rupture. Because of the failure of medical treatment on the previous ectopic pregnancy, it could be argued that surgical treatment, though more invasive, would have been the better approach. It has indeed been reported that a previous ectopic pregnancy could be a risk factor for the failure of medical therapy in interstitial pregnancies [7]. The aim of this work is to report a case of interstitial pregnancy managed in our center, where a medical approach failed and was followed by a radical surgical strategy. To date there is no gold standard regarding the management of this rare condition. Reporting different therapeutic approaches and clinical outcomes could be useful for supporting gynecologists in the choice of management techniques for interstitial pregnancies. 

## 4. Conclusions

Although medical therapy for interstitial pregnancy is indicated as the first line of treatment, the patient should be aware that it entails a higher risk of failure than surgery. On the other hand, the advantages of fertility sparing should be weighted according to the patient’s reproductive perspectives. Appropriate counselling is therefore the key in managing the treatment of interstitial pregnancy, provided that thorough medical and obstetric anamnesis is first conducted to properly illustrate the balance of risks and benefits.

## Figures and Tables

**Figure 1 medicina-58-00937-f001:**
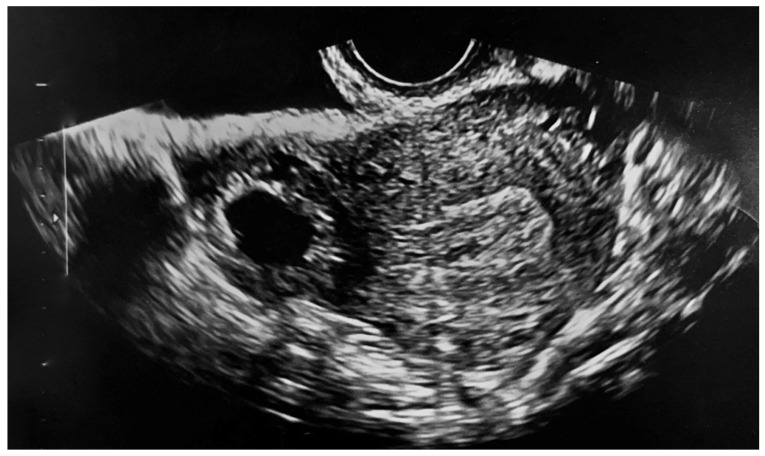
Transvaginal ultrasound showing the interstitial pregnancy.

**Figure 2 medicina-58-00937-f002:**
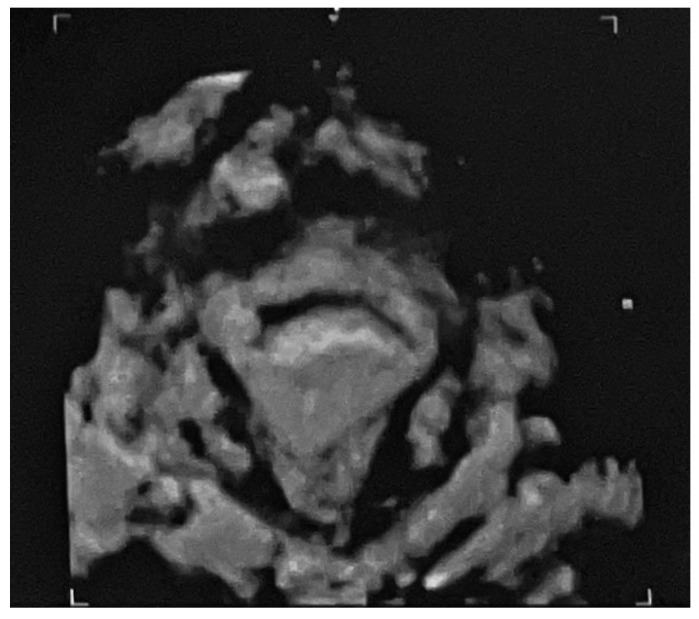
3D transvaginal ultrasound showing the interstitial pregnancy.

**Figure 3 medicina-58-00937-f003:**
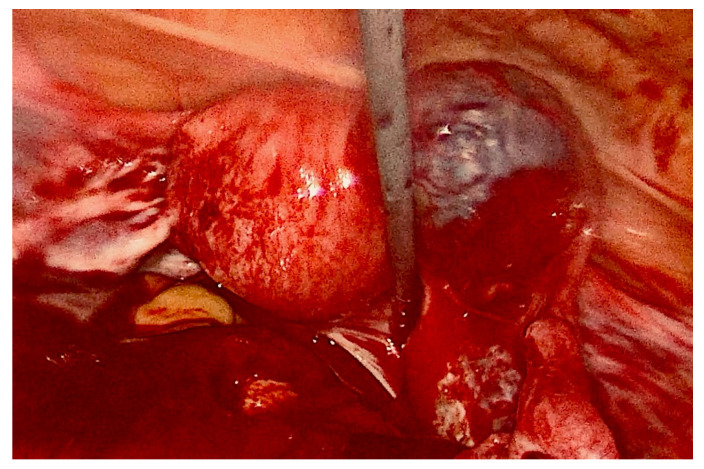
Laparoscopic view of interstitial pregnancy.

## Data Availability

Not applicable.

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
