# Peer review of "Surgical Treatment Following Failed Medical Treatment of an Interstitial Pregnancy"

_medicina, 2022, doi:10.3390/medicina58070937_

Round 1

Reviewer 1 Report

Thank you for inviting me to review this manuscript. This is a case report about an interstitial ectopic pregnancy, where the treatment with medications failed, leading to laparoscopic surgical intervention. I have some considerations before the publication of this article.

Major considerations:

1. The abstract of a case report should focus more on the case itself. It should typically include a good summary of the demographics, clinical presentation, diagnostics, therapeutic options, and prognosis. This can be followed by a quick summary of the clinical recommendations for future cases. Additionally, the authors should not forget to add a sentence about why this case is unique. Why should it be reported? What will the benefit be for doctors and patients?

2. Like any other article, an introduction of a case report should contain a statement about the knowledge gap and an aim. The introduction provided by the authors does not seem to have both.

3. What is the prognosis of the patient? Was there a follow-up? How was the follow-up planned? What further complications developed during the follow-up?

4. It is a bit bizarre that the authors are repeating the introduction in the first two paragraphs of the discussion but with some paraphrasing and little extra information. The discussion of a case report should focus mainly on what could have been done in a better way, why every decision in the management plan was taken, and how every decision could have affected the outcome of the patient.

5. The authors should discuss the strengths and limitations of their research project at the end of the discussion section.

Minor considerations:

6. For the general reader and interested medical students, the figures should stand alone in the manuscript. For example, in Figure 1, I suggest the authors provide pointers to where the uterus, gestational sac, and embryo are. Also, is there another photo with the sac diameter? There is no mention of the size of the sac in the manuscript. The same principle of pointers should be done in Figure 2 and Figure 3 as well. A final comment for the figures is that the legends should clearly describe what is seen in the photos. If the authors use pointers, the pointers should be discussed. If they use letters (e.g., A, B, C) for each structure, these should also be discussed. 

7. Were there any other relevant investigations done? If yes, please report them as well. This can be done in a table. 

8. What was the dose of leucovorin?

9. The first sentence of the discussion appears to be a typo. Please fix it.

10. I encourage the authors to use and cite the CARE checklist for reporting case reports. I also encourage them to attach the checklist as an online supplementary file. Here is the link to the checklist: https://www.equator-network.org/reporting-guidelines/care/. Nevertheless, I attached it.

Proofreading errors:

Line 37: "disomogenous, ipoechoic" should be "dishomogeneous, hypoechoic".

Line 55: "vaginal blood loss" should be "vaginal bleeding".

Line 69: the authors should use the British or American language consistently. For example, most words were used in the American language, but "counselling" with double-L is not American.

Line 72: please change "hemodynamical" to "hemodynamic".

Line 75: it should be "for all 10 days".

Line 76: ultrasound findings cannot be "stable", they can be "inconclusive" or "normal".

Line 78: it should be "beta-hCG serum [level]"

Author Response

Thank you for inviting me to review this manuscript. This is a case report about an interstitial ectopic pregnancy, where the treatment with medications failed, leading to laparoscopic surgical intervention. I have some considerations before the publication of this article.

Major considerations:

  1. The abstract of a case report should focus more on the case itself. It should typically include a good summary of the demographics, clinical presentation, diagnostics, therapeutic options, and

prognosis.

In accordance with the reviewer's suggestions, we have made the requested changes.

  1. Like any other article, an introduction of a case report should contain a statement about the knowledge gap and an aim. The introduction provided by the authors does not seem to have both. We collected the reviewer's comments and made the changes suggested
  2. What is the prognosis of the patient? Was there a follow-up? How was the follow-up planned? What further complications developed during the follow-up?

We thank the reviewers. We have made the requested changes.

  1. It is a bit bizarre that the authors are repeating the introduction in the first two paragraphs of the discussion but with some paraphrasing and little extra information. The discussion of a case report should focus mainly on what could have been done in a better way, why every decision in the management plan was taken, and how every decision could have affected the outcome of the patient

In accordance with reviewer’s suggestion, we have modified the text.

  1. The authors should discuss the strengths and limitations of their research project at the end of the discussion section

Thank you for your comment. We added some specification in discussion section.

Minor considerations:

  1. For the general reader and interested medical students, the figures should stand alone in the manuscript. For example, in Figure 1, I suggest the authors provide pointers to where the uterus, gestational sac, and embryo are. Also, is there another photo with the sac diameter? There is no mention of the size of the sac in the manuscript. The same principle of pointers should be done in Figure 2 and Figure 3 as well. A final comment for the figures is that the legends should clearly describe what is seen in the photos. If the authors use pointers, the pointers should be discussed. If they use letters (e.g., A, B, C) for each structure, these should also be discussed.

In accordance with reviewer’s suggestion, we have modified the figures.

  1. Were there any other relevant investigations done? If yes, please report them as well. This can be done in a table.

We thank the reviewers for their consideration. We believe that further investigations were not done and were not required. So we did not include an additional table.

  1. What was the dose of leucovorin?

We collected the reviewer's comments and we have added it.

  1. The first sentence of the discussion appears to be a typo. Please fix it.

Thank you for your comment. We modified it.

  1. I encourage the authors to use and cite the CARE checklist for reporting case reports. I also encourage them to attach the checklist as an online supplementary file. Here is the link to the checklist: https://www.equator-network.org/reporting-guidelines/care/.

Nevertheless, I attached it.

We collected the reviewer's comments and we have added it

Proofreading errors:

Line 37: "disomogenous, ipoechoic" should be "dishomogeneous, hypoechoic"-

Line 55: "vaginal blood loss" should be "vaginal bleeding"

Line 69: the authors should use the British or American language consistently. For example, most words were used in the American language, but "counselling" with double-L is not American.

Line 72: please change "hemodynamical" to "hemodynamic

Line 75: it should be "for all 10 days".

Line 76: ultrasound findings cannot be "stable", they can be "inconclusive" or "normal".

Line 78: it should be "beta-hCG serum [level]"

In accordance with reviewer’s suggestion, we have modified the text.

Reviewer 2 Report

This is a very interesting case where the use of MTX and mifepristone failed to suppress the ectopic pregnancy development. My main concern is the initial β-hCG level was quite high, but why did you choose MTX intramuscular injection + Mifepristone rather than intralesional MTX injection combined with systemic injection? A research by Halperin showed the high successful systemic injection treatment only in group with lower β-hCG level (DOI: 10.1159/000073774)

For the image, if it is possible, would you add a marker (arrow or asterisk) in the presence of interstitial pregnancy images (2D and 3D US)?

Author Response

This is a very interesting case where the use of MTX and mifepristone failed to suppress the ectopic pregnancy development. My main concern is the initial β-hCG level was quite high, but why did you choose MTX intramuscular injection + Mifepristone rather than intralesional MTX injection combined with systemic injection? A research by Halperin showed the high successful systemic injection treatment only in group with lower β-hCG level (DOI: 10.1159/000073774)

We collected the reviewer's comments and we have modified it

For the image, if it is possible, would you add a marker (arrow or asterisk) in the presence of interstitial pregnancy images (2D and 3D US)?

I thank the reviewer for their comment. We have made the requested changes.

Reviewer 3 Report

Nicely written case report however need to address few areas 

1: Line 88: This line needs editing

''Authors should discuss the results and how they can be i Interstitial pregnancy is a 88 rare condition, accounting for 2-4% of all tubal ectopic pregnancies.''

2: I cant find any evidence and role of mifepristone before medical management so please clarify that '' is this local experience of clinician or any guideline from any International O&G societies ; because developed countries like UK, USA and Australia will not be interested in any information which has no credible recommendation.

3: Please also mention the possibility of retained products with medical treatment which can be problematic to manage.

4 : Line 71: '' Due to the past obstetrical history, ultrasound features, hemodynamical stability and the patient’s desire of future pregnancies, the medical treatment was chosen''

Please clarify what do you mean by '' previous obstetric history'' because risk factors for ectopic in this case is not mentioned and which was the factor in obstetric history which Authors wants to talk about. 

Author Response

Nicely written case report however need to address few areas

1: Line 88: This line needs editing ''Authors should discuss the results and how they can be i

Interstitial pregnancy is a 88 rare condition, accounting for 2-4% of all tubal ectopic pregnancies.'' We collected the reviewer's comments and we have modified it

2: I cant find any evidence and role of mifepristone before medical management so please clarify that '' is this local experience of clinician or any guideline from any International O&G societies ;

because developed countries like UK, USA and Australia will not be interested in any information which has no credible recommendation.

We thank the authors for their comments. We have explained what is required in lines 139-144

3: Please also mention the possibility of retained products with medical treatment which can be problematic to manage.

In accordance with reviewer’s suggestion, we have modified the text.

4 : Line 71: '' Due to the past obstetrical history, ultrasound features, hemodynamical stability and the patient’s desire of future pregnancies, the medical treatment was chosen'' Please clarify what do you mean by '' previous obstetric history''  because risk factors for ectopic in this case is not mentioned and which was the factor in obstetric history which Authors wants to talk about.

I thank the reviewer for their comment. We have made the requested changes.

Round 2

Reviewer 1 Report

Thank you for the revision. I am requesting some minor adjustments:

Line 19, line 27: monolateral is not actually an English word. Please replace it with "unilateral".

Lines 21-24: it should read like this: We report a case of IP in a 29-year-old Caucasian woman with a history of unilateral salpingectomy for ectopic pregnancy. In this instance, she was medically treated with a double dose of intramuscular MTX 50 mg/m2 combined with a single dose of leucovorin 15 mg and a single dose of mifepristone 600 mg orally.

Line 82, line 118: please remove the word "instrumental".

Line 100: What was the follow-up plan after the second discharge? In lines 93-94, the authors prescribed weekly b-hCG and USG; was that the plan in the second discharge as well? Or was it different?

Finally, the authors' contributions are a bit confusing. Please re-write this part to make it clear who did what.

Author Response

Thank you for the revision. I am requesting some minor adjustments:

Line 19, line 27: monolateral is not actually an English word. Please replace it with "unilateral".

Lines 21-24: it should read like this: We report a case of IP in a 29-year-old Caucasian woman with a history of unilateral salpingectomy for ectopic pregnancy. In this instance, she was medically treated with a double dose of intramuscular MTX 50 mg/m2 combined with a single dose of leucovorin 15 mg and a single dose of mifepristone 600 mg orally.

Line 82, line 118: please remove the word "instrumental".

Line 100: What was the follow-up plan after the second discharge? In lines 93-94, the authors prescribed weekly b-hCG and USG; was that the plan in the second discharge as well? Or was it different?

Finally, the authors' contributions are a bit confusing. Please re-write this part to make it clear who did what.

In accordance with reviewer’s suggestion, we have modified the text